Bimetallic nanoparticles and biochar produced by Adansonia Digitata shell and their effect against tomato pathogenic fungi

Aldahasi Reham M.
http://orcid.org/0000-0002-5336-038X Shami Ashwag
Mohammed Afrah E. AFAMohammed@pnu.edu.sa
Department of Biology, College of Science, Princess Nourah bint Abdulrahman University , Riyadh , Saudi Arabia
Mora-Montes Héctor
Electronic publication date: 2024 Mar 1
Publication date: 2024
Volume: 12
Electronic Location ID: e17023
Received 2023 Oct 20; Accepted 2024 Feb 6
Copyright: © 2024 Aldahasi et al.
Copyright year: 2024
Copyright holder: Aldahasi et al.
License: This is an open access article distributed under the terms of the Creative Commons Attribution License, which permits unrestricted use, distribution, reproduction and adaptation in any medium and for any purpose provided that it is properly attributed. For attribution, the original author(s), title, publication source (PeerJ) and either DOI or URL of the article must be cited.
License URL: https://creativecommons.org/licenses/by/4.0/

Keywords: Silver nanoparticles, Fusarium sp., Iron oxide nanoparticles, Sclerotinia sclerotiorum, Alternaria sp., Bimetallic nanoparticles, Antifungal activity

Funding: Deanship of Scientific Research at Princess Nourah bint Abdulrahman University, through the Research Groups Program RI-44-0768 This work was funded by the Deanship of Scientific Research at Princess Nourah bint Abdulrahman University, through the Research Groups Program Grant no. (RI-44-0768). The funders had no role in study design, data collection and analysis, decision to publish, or preparation of the manuscript.

==============================
Adansonia digitata L. is a royal tree that is highly valued in Africa for its medicinal and nutritional properties. The objective of this study was to use its fruit shell extract to develop new, powerful mono and bimetallic nanoparticles (NPs) and biochar (BC) using an eco-friendly approach. Silver (Ag), iron oxide (FeO), the bimetallic Ag-FeO NPs, as well as (BC) were fabricated by A. digitata fruit shell extract through a reduction process and biomass pyrolysis, respectively, and their activity against tomato pathogenic fungi Alternaria sp., Sclerotinia sclerotiorum, Fusarium equiseti, and Fusarium venenatum were detected by agar dilution method. The Ag, FeO, Ag-FeONPs, and BC were characterized using a range of powerful analytical techniques such as ultraviolet–visible (UV–Vis) spectroscopy, scanning electron microscopy (SEM), transmission electron microscopy (TEM), Fourier Transform-Infra Red (FT-IR), dynamic light scatter (DLS), and zeta potential analysis. The fabricated Ag, FeO and Ag-FeO NPs have demonstrated a remarkable level of effectiveness in combating fungal strains. UV–Vis spectra ofAg, FeO, Ag-FeONPs, and BC show broad exhibits peaks at 338, 352, 418, and 480 nm, respectively. The monometallic, bimetallic NPs, and biochar have indicated the presence in various forms mostly in Spherical-shaped. Their size varied from 102.3 to 183.5 nm and the corresponding FTIR spectra suggested that the specific organic functional groups from the plant extract played a significant role in the bio-reduction process. Ag and Ag-FeO NPs exhibited excellent antifungal activity against pathogenic fungi Alternaria sp., S. sclerotiorum, F. equiseti, and F. venenatum. The current study could be a significant achievement in the field of antifungal agents since has the potential to develop new approaches for treating fungal infections.

Introduction

Over the past few years, there has been a significant increase in the number of emergent plant pathogens, which pose a threat to the stability of agroecosystems and the preservation of native biodiversity (Corredor-Moreno & Saunders, 2020). There are several kinds of plant pathogens, including fungi, bacteria, viruses, and nematodes, that can cause severe damage to crop worldwide. These pathogens do not just result in significant losses but also decrease the quality and quantity of agricultural products (Thambugala et al., 2020). Furthermore, if a pathogen infects crops in the field or during post-harvest storage, it can harm the health of both humans and livestock. Tomatoes (Lycopersicon esculentum Mill) a member of the Solanaceae family are an essential vegetable globally (Mostafa et al., 2021). Additionally, tomato products are known for their significant nutritional value as well as their antioxidant, anti-inflammatory, and anticancer properties (Salehi et al., 2019) however, suffer greatly from various diseases such as early blight, caused by Alternaria solani which led to 80% loss in tomato yield production (Pandey et al., 2020). Sclerotinia stem rot, caused by Sclerotinia sclerotiorum, is another serious disease that can have detrimental effects on tomato (Mazumdar, 2021). Fusarium species are also notorious for their negative impact on agricultural food production, inducing decreased crop yields and economic losses (Akpinar, Unal & Sar, 2021). Fusarium equiseti is extremely destructive to tomato crops, causing Fusarium wilt disease (El-Nagar et al., 2022). Such fungal infections might be treated by chemical fungicides which could be a solution for a long time, but their use is always associated fungicide residues which lead to environmental pollution and increase pathogen resistance to fungicides (Zhang et al., 2020; Salazar et al., 2022). Carbendazim, a systemic fungicide, has widespread use in the agricultural sector for managing plant fungal diseases such as blight disease in tomatoes caused by Alternaria alternata (Singh et al., 2016). However, its excessive use has led environment pollution, resulting in potential human health hazard (Jewaliya et al., 2021; Zhou et al., 2023). Generally, the usage of chemically synthesized pesticides is being reduced as world trends shift and various strategies are used in plant disease management (Thambugala et al., 2020). Therefore, it is highly warranted to develop sustainable techniques and tools bypassing traditional agriculture practices. Nanotechnology offers a simple, lucid solution to the problems in disease management (Alghuthaymi et al., 2021). Numerous beneficial applications in biotechnology and microbiology have been yielded by nanotechnology (Alherz et al., 2022). Nanoparticles possess exceptional thermal conductivity, catalytic reactivity, non-linear optical performance, and chemical steadiness due to their large surface area to volume ratio. These small fragments have a nano-scale dimension that ranges between 1–100 nm (Agarwal, Venkat Kumar & Rajeshkumar, 2017). Moreover, using nanoparticles that fabricated through physical and chemical pathways can be expensive and may pose environmental hazards (Khan et al., 2022). Therefore, development of an eco-friendly and cost-effective method is needed for nanoparticles synthesis (Singh et al., 2020). Green synthesis technology is biologically safe, reliable, and nontoxic method to synthesize nanoparticles using microorganisms and plants (Dikshit et al., 2021) however, researchers have recently shifted their focus toward synthesizing metal nanoparticles using plant extracts (Khan et al., 2022). In addition, bimetallic nanoparticles (BNPs) represent a highly intricate nanoscale combination of two distinct metal constituents. It is a well-established fact that the exceptional properties exhibited by BNPs far surpass those of monometallic nanoparticles, making them the subject of great interest from both scientific and technological perspectives. Unique mixing patterns and synergistic effects render bimetallic nanoparticles an even more attractive option when compared to monometallic nanoparticles (Dlamini, Basson & Pullabhotla, 2023).

In the current study more focus was applied on the fabrication of silver, and iron in both mono and bimetallic nanoparticles forms beside the biochar as agents to control the phytopathogen. Biochar (BC) is a solid substance rich in carbon that is formed through the pyrolysis of organic substances under conditions of low or limited oxygen (Feng et al., 2021). This substance’s unique properties, including high carbon content, cation exchange capacity, large specific surface area, and stable structure, have drawn attention. Its high aromaticity and resistance to decomposition make it a versatile resource in agriculture, environment, and energy (Wang & Wang, 2019; Feng et al., 2021). The extensive application of silver nanoparticles (AgNP) in various industrial sectors is primarily due to their highly effective antibiotic properties. Previous studies have shown the stability and efficiency of AgNPs mediated by Polyalthia longifolia and Polygonatum geminiflorum against F. oxysporum (Ahmad et al., 2022; Dashora et al., 2022). In addition, the magnetic nature of iron oxide nanoparticles (FeONPs) has garnered significant attention in recent years. FeONPs synthesized using Euphorbia hirta leaf extract exhibited potent and significant antifungal activity against Anthrogrophis cuboida, Aspergillus fumigatus, and Aspergillus niger (Ahmad, Kumar Jaiswal & Amjad, 2021). The Gardenia jasminoides synthesized Ag-Fe BNPs that demonstrated a superior antimicrobial effect in comparison to the monometallic nanoparticles against Candida albicans yeast (Padilla-Cruz et al., 2021). Further, the combination of poultry feces and sawdust wastes was used to produce biochar has been proven to be highly effective in preventing the growth of the pathogenic Fusarium verticillioides, which is responsible for ear rot in maize (Akanmu et al., 2020).

The target plant for the current study which used as a biogenic agent in nanoparticle fabrication was the baobab (Adansonia digitata L.), is a remarkable and colossal tree that is indigenous to several African nations (Kabbashi et al., 2017). Its medicinal and nutritional benefits are widely assessed (Kamatou, Vermaak & Viljoen, 2011). Furthermore, it is imperative to note that baobab is rich in phytochemicals such as flavonoids, phytosterols, amino acids, fatty acids, vitamins, and minerals (Rahul et al., 2015). This substance possesses a multitude of biological properties, such as antimicrobial, anti-malarial, diarrhoeal, anemic, asthmatic, antiviral, antioxidant, and anti-inflammatory activities, among others (Kamatou, Vermaak & Viljoen, 2011; Rahul et al., 2015). It is worth noting that no study has previously reported the biosynthesis of NPs using Adansonia digitata fruit shell therefore, the main objective of this study was to produce AgNPs, FeONPs, Ag-FeONPs, and biochar (BC) using Adansonia digitata fruit shell as a capping agent. Additionally, the study aimed to test the biological activity of these biofabricated materials against phytopathogen fungi Alternaria sp., S. sclerotiorum, F. equiseti and F. venenatum.

Materials and Methods

Chemical and reagents

Silver nitrate, ferrous chloride hydrated, and ferric chloride (hexahydrate) pure and potato-dextrose agar (PDA) were obtained from Laboratory of Princess Nourah bint Abdulrahman University, Riyadh, Saudi Arabia Alternaria sp. (ON876488), F. venenatum (ON876497), S. sclerotiorum (ON876490) and F. equiseti (ON876498) have been isolated from infected tomato (Solanum lycopersicum) at Health Sciences Research Center at Princess Nourah bint Abdulrahman University and Adansonia digitata fruit were purchased from Sudan market. The commercial fungicide Tebusha 250 EW (Sharda Cropchem, Spain) containing Tebuconazole 25%.

Preparation of aqueous extract

Fruit shells of Adansonia digitata were washed with distilled water to remove any dust particles and other contaminants, then dried overnight at 70 °C and milled to a fine powder, which was used for further processing. An aqueous extract of Adansonia digitata fruit shell was prepared by adding 5 g to 100 mL of distilled water and heated for 15 min at 90 °C. After that, the extract was filtered through filter paper with a pore diameter of 20 μm.

Synthesis of nanoparticles

AgNPs were synthesized by adding 90 mL of 1 mM aqueous AgNO3 solution to 10 mL of plant extract in an Erlemeyer flask and heated for 15 min at 90 °C. FeONPs were prepared by adding 50 mL of a solution composed of FeCl2 and FeCl3 (1:2) at a concentrations of 1 mM to 50 mL of plant extract in a 1:1 ratio in an Erlemeyer flask and heated for 15 min at 90 °C. The formation of nanoparticles was initially detected by the change in color. After being cooled to room temperature, the solutions were centrifuged at 5,580 × g for 30 min. then washed twice with distilled water, and the resulting pellet was dried in an oven for 10 min at 45 °C. Finally, NPs were stored for further study.

Synthesis of bimetallic nanoparticles

AgNO3 (1.6 g) was dissolved in 100 mL of plant extract (14.2 g in 100 mL of distilled water) at 70 °C. Once the salt had been fully dissolved, (0.3 g) of FeCl2 and (0.6 g) of FeCl3 were added to the mixture. The mixture was then heated at 90 °C for an hour. Afterward, the nanoparticles were centrifuged at 5,580 × g for 1 h. Then it was washed with distilled water and the pellet was dried in an oven at 45 °C for 10 min. Finally, the synthesized NP powder was collected for further study.

Preparation of biochar (BC)

Biochar was produced by pyrolysis process using a muffle furnace under limited oxygen conditions, 18 g from the fruit shells of Adansonia Digitata, and promptly placed in a crucible, which was then heated in the muffle furnace at a temperature of 500 °C for 3 h. It was left to cool overnight at room temperature. Then, the biochar is ready for study.

Characterization of metal, bimetallic nanoparticles, and biochar

Various techniques were employed to determine the properties of the metal, bimetallic nanoparticles, and biochar produced in this investigation.

Ultraviolet-visible spectroscopy

UV–Vis spectral analysis was conducted using an Evolution 201 UV-Visible spectrophotometer (Thermo Fisher Scientific, Waltham, MA, USA). The reaction mixture was tested within a range of 200 to 500 nm after 24 h, with distilled water acting as a blank.

Fourier-transform infrared spectroscopy

FTIR spectroscopy (SPECTRUM100; Perkin-Elmer, Waltham, MA, USA) analyzed the functional groups in the phytoconstituents that are responsible for reducing and capping nanoparticles. The scanning range of FTIR spectroscopy was between 500–4,000 cm.

Dynamic light scattering (DLS) and zeta potential

Measurements of nanoparticle hydrodynamic particle size distribution, PDI, and zeta potential were conducted using DLS with Zetasizer (NANO ZSP; Malvern Instruments Ltd., Serial Number: MAL1118778, ver 7.11, Malvern, UK).

Scanning electron microscopy (SEM) and transmission electron microscopy (TEM)

SEM (JEOL, Tokyo, Japan) integrated with energy dispersive X-ray (EDX) and elemental mapping was used to examine the surface morphology, size, and element composition of metals in biosynthesized nanoparticles. TEM (JEOL, Tokyo, Japan) was used to determine the size and morphology of nanoparticles.

In-vitro antifungal assay

Biosynthesized AgNPs, FeONPs, BNPs, and the biochar were tested as antifungal agents against the tomato pathogen fungi Alternaria sp., S. sclerotiorum, F. equiseti, and F. venenatum using the agar dilution method. Concentrations of 10 mg/mL were prepared from AgNPs, FeONPs, Ag-FeONPs, and BC then each concentration was added into a Petri plate then 9 ml of sterilized potato dextrose agar was added before solidification as previously described by Khatami et al. (2018). The negative control was the PDA media without treatment and the positive control was the PDA media inoculated with fungicide Tebuconazole at 10 ppm concentration. An inoculum of 9 mm diameter of each fungal strain was taken from a 7-day-old culture and then placed aseptically at the center of the solidified agar, and the plates were kept in an incubator at 25 °C for 4 days.

Microscopic observation of fungal growth

To investigate the impact of treatments on the fungal growth, FeONPs was chosen as a candidate to verify the possible changes of spores and mycelial growth. The cells from treated and control plates were collected using a sterilized loop from the surface of petri plates that contain Alternaria sp., S. sclerotiorum, F. equiseti, and F. venenatum. The morphology of the fungal cells was observed under a light microscope (SLA2000; LABOMED, Los Angeles, CA, US).

Statistical analysis

The statistical analysis was obtained by analyzing analysis of variance (two-way ANOVA) to determine significant differences (P ≤ 0.001) among the study factors besides Least significant differences. The statistical analysis was performed using GraphPad Prism version 10.0.2.232 (GraphPad Software, La Jolla, CA, USA). The spectra for FTIR and UV-Vis were produced using OriginPro® 2023b.

Results

The current study employed A. digitata fruit shell extract as a powerful reducing agent to synthesize mono and bimetallic NPs and biochar as an eco-friendly and sustainable approach for nanoparticle fabrication. Additionally, the synthesized NPs and biochar (BC) underwent comprehensive characterization through various analytical and spectroscopic techniques and then their antifungal activity against Alternaria sp., S. sclerotiorum, F. equiseti, and F. venenatum has been investigated.

Ultraviolet-visible spectroscopy

Before measuring the UV absorbance, changes in the color of the reaction medium (A. digitata fruit shell extract + individual metal salt) were observed, indicating the formation of the NPs. UV-visible spectroscopy absorbance peaks at wavelengths between 200 to 500 nm for the plant extract and phytofabricated NPs are displayed in Figs. 1A–1E. The absorption peak of the A. digitata fruit shell extract was observed at 303 nm which is specific to flavonoid absorptions. Which indicated by Braca et al. (2018) when studied the primary constituents of A.digitata extract are flavonoids, including flavan-3-ols, flavonols, tannins, and flavones. The flavonoids present in the A. digitata extract are responsible for reducing the metal solution to nanoparticles and stabilizing the solution. AgNPs showed an absorption peak at 418 nm, FeONPs at 338 nm, Ag-FeO NP at 352 nm and the biochar at 480 nm. Similar findings were also reported since peak at 418 nm was observed for AgNPs fabricated by Leonotis nepetifolia (Nagaraja et al., 2022). FeONPs synthesized using leaf extract of Euphorbia herita demonstrated peak at 345 nm (Ahmad, Kumar Jaiswal & Amjad, 2021). A peak at 350 nm was noted for the Ag-FeO NP synthesized by Gardenia jasminoides (Padilla-Cruz et al., 2021). In addition, the biochar derived from maize straw exhibited a peak at 450 nm (Kamal et al., 2022). The current findings indicated that A. digitata fruit shell extract was successful in fabricating the mono and bimetallic NPs and the biochar since the UV peaks reported were between 350 to 500 nm.

Figure 1 UV-vis spectra showing absorbance peaks of A. digitata fruit shell extract (A), AgNPs (B), FeONPs (C), Ag-FeONPs (D), and biochar (E) prepared by the fruit shell extract.

Fourier-transform infrared spectroscopy

A. digitata’s fruit shell contains several secondary metabolites, including phenolic acids, flavonoids, organic acids, hydroxy fatty acids, and saponin (Ismail et al., 2019). These compounds are vital in the reduction process and have a capping effect, which leads to the creation of nanoparticles (Marslin et al., 2018). To detect any changes in the bonding that appear during the metal reduction and nanoparticle formation, FT-IR measurements were conducted for both A. digitata fruit shell extract and the NPs reduced by its aid as shown in Fig. 2. The FTIR spectrum of A. digitata fruit shell extract was noted at the following peaks: 3,327 cm−1 that corresponds to O-H (hydroxyl) stretching from the secondary amine (Eze & Nwaeze, 2022), 2,027 cm−1 assigned to the C-H (alkane) stretching vibrations in aromatic compound (Akintola et al., 2020); and 1,639 cm−1 related to the N-H bond of primary amine (Muthukumar et al., 2020). The FTIR spectra for AgNPs, FeONPs, Ag-FeONPs, and BC are shown in Fig. 2 which displayed the absorption peaks at 3,327, 2,027, and 1,639 cm−1. These observations confirmed that the synthesized AgNPs, FeONPs, Ag-FeONPs, and BC were capped with the secondary metabolites from the A. digitata fruit shell extract. It is essential to note that these components are necessary for preventing the aggregation of nanoparticles and are critical in stabilizing their structure. Recent research about AgNPs fabricated using Morus indica extract exhibited peaks at 1,630 and 3,421 cm−1 (Some et al., 2019). FeONPs synthesized using Mikania mikrantha extract revealed bands at 3,302 and 3,317 cm−1, as well as 1,636 cm−1 (Biswas et al., 2020). The Ag-FeONPs synthesized using Salvia Officinalis extract exhibit FTIR spectra at 3,309 cm−1 (Malik, Alshehri & Patel, 2021). Additionally, a wide band at 3,254 cm−1 was noted for the biochar from plantain fibers (Adeniyi, Ighalo & Onifade, 2021).

Figure 2 The FTIR spectra of A. digitata fruit shell extract, phyto-fabricated nanoparticles, and biochar.

Dynamic light scattering (DLS) and zeta potential

Dynamic light scattering (DLS) technique was utilized to detect the synthesized nanoparticles’ hydrodynamic size, polydispersity indices (PDI), and surface zeta potential (Nayak et al., 2016). The AgNPs had a size of 102.3 nm as shown in Fig. 3A, similar range of observation was noted for AgNPs fabricated by Angelica gigas that showed an average size of 102 nm (Ryu, Nam & Baek, 2022). Figure 4A displayed an average size of 183.5 nm for FeONPs, a similar finding was reported by Umair (2022) who applied Elwendia persicum for FeONPs fabrication which showed an average size of 194.5 nm. The Ag-FeO NP’s mean size distribution was 176.2 nm, as shown in Fig. 5A. The PDIs for AgNPs, FeONPs, and Ag-FeONPs were 0.25, 0.22, and 0.08, respectively. Such findings indicated that the NPs fabricated in the current study were relatively uniform in their size distribution since the PDI values were below 0.5 (Danaei et al., 2018). Zeta potentials for AgNPs, FeONPs, Ag-FeONPs, and the biochar were −32.72, −31.78, −42.8, and −27.09 mV, respectively as shown in Figs. 3B, 4B, 5B and 6. The negative zeta potential indicates that the nanoparticles were stable (Kumavat & Mishra, 2021) and the strong negative charges prevent the particles from being agglomerated (Bhagat et al., 2019). Such negative values might be from the phytochemicals indicating their role as capping and stabilizing effect. Negative zeta potentials were also reported at −33.5 mV for AgNPs mediated by Atropa acuminata (Rajput, Kumar & Agrawal, 2020), −25.2 mV for FeONPs produced by tea waste (Periakaruppan et al., 2021), −24.3 mV was reported for the biochar produced from by Persicaria salicifolia (Hosny, Fawzy & Eltaweil, 2022).

Figure 3 (A) Size distribution and (B) zeta potential distribution for the AgNPs fabricated by A.digitata.

Figure 4 (A) Size distribution and (B) zeta potential distribution for FeONPs fabricated by A.digitata.

Figure 5 (A) Size distribution and (B) zeta potential distribution for the bimetallic Ag-FeONPs fabricated by A. digitata.

Figure 6 Zeta potential distribution for Biochar derived from A. digitata fruit shell.

Scanning electron microscopy (SEM) and transmission electron microscopy (TEM)

SEM analysis has been widely used to analyze the morphological surface of AgNPs, FeONPs, and bimetallic NPs besides the biochar and detecting its macropores. Moreover, EDX was employed to identify the elemental composition of tested materials accurately. In Fig. 7A, indicated the morphology of AgNPs that was a mixture of cylindrical, cubic, spherical, and triangular shapes with a smooth surface where no agglomeration was noted. EDX spectrum clearly showed peaks at 0.1 and 3 keV, indicating the presence of Ag (Fig. 7B). Figure 7C displayed the EDX mapping analysis of elements, and it indicates the presence of Ag, carbon and oxygen. Similarly, AgNPs synthesized using Syngonium podophyllum leaf extract showed a rectangle, rod, triangle, and spherical shapes (Naaz et al., 2021). Tran et al. (2021) showed the presence of the element signal of Ag at 3 keV. Furthermore, Fig. 8A showed irregular spherical shape for FeONPs with smooth surfaces and Fe ion peaks were noted at 0.8 (strong peak) and 6.2 keV (slight peak) and the distribution of FeONPs beside C and O were noted in Figs. 8B and 8C. Similar findings was reported by Chawla et al. (2021) when applying the Gum Arabic to fabricate Fe(OH)₂ that demonstrated the elemental iron peak at 0.8 and 6.3 keV. The SEM image were indicated in Fig. 9A for the bimetallic Ag-FeONPs that relatively spherical in shapes where Fig. 9B, confirm the presence of both Ag and Fe nanoparticles and peaks detected at 0.1, 2.8, and 3.1 keV confirmed the presence of Ag, while those observed at 0.8, 6.3, and 7 keV identified Fe. The elemental mapping showed that the yellowish-green color of the Ag-FeONPs sample was due to the presence of FeONPs, while the green coloration was related to AgNPs (Fig. 9C). Spherical shape was noted for Ag-Au NPs fabricated by Hippeastrum hybridum (Sher et al., 2022). Similarly, peaks appeared at 3 (Ag), 6 (Fe), and 0.9 keV (Fe) were reported for Ag-FeONPs prepared by Salvia officinalis (Malik, Alshehri & Patel, 2021). The SEM images of the biochar is shown in Fig. 10 which indicated a rough surface with a vessel structure which mostly are irregular spherical and cylindrical shapes. The EDS spectrums in Fig. 10B displayed a sharp peak of carbon at 0.1 keV and a slight peak of oxygen at 0.2 keV, indicating the presence of these elements in the biochar’s surface chemical composition which also displayed in the elemental mapping in Fig. 10C. Recent studies have shown that biochar prepared from Fenton sludge has a rough surface and an irregular flaky structure (Liu et al., 2022). On the other hand, biochar derived from rubberwood sawdust had a cylindrical shape (Ali et al., 2022). The detected C and O could be attributed to the phytochemical from A. digitata fruit shell that may cap the nanoparticles.

Figure 7 (A) SEM image; (B) EDX pattern and elemental composition; (C) element mapping of AgNPs.

Figure 8 (A) SEM image; (B) EDX pattern and elemental composition; (C) element mapping of FeONPs.

Figure 9 (A) SEM image; (B) EDX pattern and elemental composition; (C) element mapping of Ag-FeONPs.

Figure 10 (A) SEM image; (B) EDX pattern and elemental composition; (C) element mapping of biochar.

On the other hand, TEM was used for morphology and size analysis of mono and bimetallic NPs and Biochar. Similar to SEM results, AgNPs had a spherical shape with an average size of 29.7 nm (Fig. 11A). FeONPs had a spherical shape and average size of 17.02 nm (Fig. 11B). Ag-FeO nanoparticles had a spherical shape with an average size of precisely 30.21 nm (Fig. 11C). Finally, the biochar particles exhibited a unique and complex appearance, with a round shape (Fig. 11D). TEM analysis was used to describe the Citrus reticulata blanco peel extract mediated AgNPs that resulting in spherical particles with an average size range of 10–30 nm (Jaast & Grewal, 2021). Similarly, synthesized FeONPs by Chromolaena odorata extract showed spherical particles with average size in the range of 5.6–16.8 nm (Nnadozie & Ajibade, 2020). Tea powder was also found to be effective in synthesizing spherical shape bimetallic Ag-AuNPs (Lagashetty et al., 2021). Finally, Salvia officinalis extract was used to synthesize Ag-FeO NPs, which were also spherical with an average particle size of 27.48 nm (Malik, Alshehri & Patel, 2021). The size of the nanoparticles obtained by TEM is lower that obtained by zeta size analysis, which might be due to differences in the sample sizes and the principles of the technical methods used.

Figure 11 The TEM images of (A) AgNPs, (B) FeONPs (C) Ag-FeONPs (D) biochar.

Antifungal ability of the fabricated materials

The antifungal activity of phyto-synthesized Ag, FeO, and Ag-FeO NPs and the biochar were evaluated against tomato pathogenic fungi and distinctly illustrated in Figs. 12 and S1. The A. digitata-based AgNPs exhibited a highly significant impact against Alternaria sp, S. sclerotiorum, F. venenatum, and F. equiseti (P < 0.0001). The statistical analysis demonstrated significant variations among the fungal strains, treatments, and their interaction, with a P-value (<0.0001).

Figure 12 Antifungal activity of AgNPs, FeONPs, bimetallic NPs, and biochar against all tested fungi.

Generally, AgNPs were able to suppress the growth of Alternaria sp, S. sclerotiorum, and F. venenatum by 100%, and F. equiseti by 37%.

Recent studies have demonstrated that AgNPs fabricated using pomegranate peel, Trichoderma harzianum, Grass waste, and strawberry waste were effective against A. solani, S. sclerotiorum, F. solani, and F. oxysporum, respectively (Khatami et al., 2018; Khan et al., 2021; Mostafa et al., 2021; El-Ashmony et al., 2022). A. digitata based-FeONPs were effective against Alternaria sp, S. sclerotiorum, F. venenatum, and F. equiseti at (P < 0.0001) except against Fusarium sp. where lower than that of AgNPs at the same concentration was noted.

FeONPs showed Bimetallic NPs, suppression ability, with rates of 13%, 89%, 61%, and 30% for Alternaria sp, S. sclerotiorum, F. venenatum, and F. equiseti, respectively.

Recently FeONPs fabricated by Laurus nobilis, Trichoderma harzianum, and microalga Chlorella K01 were effective against A. alternata, S. sclerotiorum, and Fusarium sp., respectively (Bilesky-José et al., 2021; Win et al., 2021; Yassin et al., 2023). The bimetallic Ag-FeONPs have demonstrated efficacy in combating fungal pathogens with great success (P < 0.0001). It is widely known that AgNPs and FeONPs retain their properties even when they are integrated into a nanocomposite (Panáček et al., 2006; Prucek et al., 2011). Generally, the noted activity of the bimetallic Ag-Fe NPs was higher than that for each metal alone since the suppression rate of the phytopathogens was 100%.

A recent finding has demonstrated that Gardenia jasminoides has the capability to fabricate effective Ag-FeONPs against C. albicans (Padilla-Cruz et al., 2021). Furthermore, zinc oxide-iron nanoparticles fabricated mushroom extract and Ag-Au NPs produced by Hippeastrum hybridum extract were effective in combating Aspergillus sp. (Sher et al., 2022; Kamal et al., 2023). Ag@ZnO core–shell nanocomposites showed antifungal activity against C. krusei at 250 μg/mL, Das et al. (2016).

There was no activity for the biochar against the tested fungi was noted. Similar findings were also reported for the biochar derived from apricot seed and olive seed against A. niger, C. parasitica, P. cinnamomic, and P. tracheiphilus (Yıldızlı, Coral & Ayaz, 2021). However, sewage sludge-derived biochar at a temperature of 500 °C indicated antifungal against the phytopathogenic fungi M. phaseolina, S. sclerotiorum, S. rolfsii, S. cepivorum, R. solani however F. oxysporum only suppressed by the same biochar but prepared at 300 °C (de Araujo et al., 2021). Such observations suggested that the antifungal activity of the biochar is highly influenced by fungal strain as well as the preparation temperature. Microorganisms require a source of carbon and mineral nutrients, that may directly affect their growth (Dai et al., 2021). The biochar promotes the growth of a symbiotic microbial community in both the soil and the root zone (Lehmann et al., 2011). Further, the presence of Tebuconazole at 10 ppm significantly suppressed the growth of Alternaria sp, S. sclerotiorum, F. venenatum, and F. equiseti with suppression rate of 100%. Previous findings indicated significant inhibitory effects of Tebuconazole against Alternaria solani, Fusarium oxysporum f. sp. cicero, and Sclerotinia sclerotiorum (Golakiya, Bhimani & Akbari, 2018; Goswami, Tewari & Upadhyay, 2020; Dhaka & Choudhary, 2022). Our findings might provide real value of the prepared NPs since some of them had the same effect against the tested fungi, which could suggest other trials to validate their practical applications.

AgNPs prepared Sisymbrium irio L. seed extract demonstrated significant antifungal activity against A. alternata, A. brassicae, F. solani, and F. oxysporum, with efficacy rates of 92%, 50%, 67%, and 78% of the carbendazim activity, respectively (Rizwana et al., 2022). There are several hypotheses concerning the mechanisms by which NPs exhibit antimicrobial properties, however, their precise mechanisms remain unknown. The antimicrobial mechanisms of metal nanoparticles depend on various factors such as their size, charge, and morphology (Cruz-Luna et al., 2021; Rana et al., 2023). Nanoparticles act by physically interacting with the fungal cell wall, causing structural damage, and ultimately resulting in cell death (Nguyen et al., 2022) Recent studies have suggested the mechanism of the NPs as antifungal agents which can be categorized into several distinct phases. (A) NPs attach to the cell walls of fungi and then enter the fungal cell through several pathways, such as passive diffusion, endocytosis, or membrane disruption. (B) Once NPs have penetrated the fungal cell, they can be distributed to various intracellular locations. They can be found in the cytoplasm, mitochondria, endoplasmic reticulum, or nucleus. (C) When NPs enter specific compartments inside the cell, they can interact with various biomolecules like proteins, lipids, or nucleic acids and cause disruptions in their function. The interaction of NPs with fungal cells can trigger a range of cellular reactions, including oxidative stress, DNA damage, apoptosis, or autophagy, ultimately leading to cell death (Li et al., 2022; Gurunathan, Lee & Kim, 2022; Rana et al., 2023).

Morphological changes of tested fungi

Due to their low suppression ability, the impact of FeONPs on the growth of the tested fungal strains was studied under the light microscope in atrial to find out any morphological changes in spore and fungal mycelia. A remarkable decrease in mycelial growth were observed (Fig. 13). The untreated Alternaria sp. had high intensity of spore and fungal mycelia that identifiable by its light brown septate hyphae and oval to ellipsoidal conidia of various sizes (Fig. 13A). However, when treated with FeONPs, the mycelia appeared dark brown and shorter with pointed ends (Fig. 13B). Recently, the treatment with AgNPs caused damage to A. solani hyphae, leading to a loss of definition and turgor (Vera-Reyes et al., 2022). Thickened and septate hyphae of S. sclerotiorum mycelium is displayed at Fig. 13C However thin, deformed, and lysed hyphae were observed from treated plates (Fig. 13D). Biogenic AgNPs caused small fragments and damage of hyphae of S. sclerotiorum (Tomah et al., 2020). F. equiseti hyphae appeared ellipsoidal to cylindrical, straight, or curved, born on short philaids, hyaline, and septate (Fig. 13E). But the treated F. equiseti indicated macroconidia at the heads on long monophialides and oval and obovoid microconidia with a truncate base (Fig. 13F). Further, the hyphae of treated F. venenatum appeared sickle-shaped, unlike the regular, tall and defined septa as shown in 14 h and g, respectively. AgNPs cause abnormal structural formation in F. graminearum hypha (Ibrahim et al., 2020). Current microscopic examination indicated fungal final life cycle stage due to toxic environments that do not support optimal growth (Ahmad et al., 2020).

Figure 13 Morphological changes of tested fungi.

Conclusions

In this study, phytochemical-induced fabrication of AgNPs, FeONPs, Ag-FeO BNPs, and biochar BC were derived from A. digitata fruit shell. The characterization of these NPs and biochar was performed using a range of techniques including UV-vis spectroscopy, FTIR, SEM, TEM, and DLS. The SEM analysis confirmed that the formation of monometallic and bimetallic nanoparticles exhibited a range of shapes, including spherical, triangular, cubic, and cylindrical with average particle sizes for the AgNPs, FeNPs, and Ag-FeONPs at 102.3, 183, and 176 nm, respectively Ag and Ag-FeONPs exhibited a broad spectrum of activity against the tested fungi which might be related to their smaller average size diameter compared to FeNPs. Therefore, NPs may hold tremendous promise for the development of effective treatments for fungal infections that could be a viable solution to control fungal plant pathogens via safe and sustainable practices.

Supplemental Information

Supplemental Information 1 Raw data related to FTIR (figure 2).

Supplemental Information 2 Antifungal data.

Supplemental Information 3 LSD analysis.

Supplemental Information 4 Supplementary figures and table.

The authors are grateful to Monerah Almusa, Department of Biology, Faculty of Science and Dhuha Alswaid, Health Science Research Center at Princess Nourah bint Abdulrahman University for providing material reagents and facilitating the laboratory work.

Additional Information and Declarations

Competing Interests

Author Contributions

Data Availability

The authors declare that they have no competing interests.

Reham M. Aldahasi conceived and designed the experiments, performed the experiments, analyzed the data, prepared figures and/or tables, authored or reviewed drafts of the article, and approved the final draft.

Ashwag Shami conceived and designed the experiments, authored or reviewed drafts of the article, and approved the final draft.

Afrah E. Mohammed conceived and designed the experiments, authored or reviewed drafts of the article, and approved the final draft.

The following information was supplied regarding data availability:

The raw data are available in the Supplemental Files.

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
