# Peer review of "Bimetallic nanoparticles and biochar produced by Adansonia Digitata shell and their effect against tomato pathogenic fungi"

_PeerJ, doi:10.7717/peerj.17023_

## Round 0.1 · original submission · Major Revisions

Two experts have assessed this manuscript and found the content relevant. In parallel, they also raised concerns that need additional experimentation to address them. In particular, they suggest including more evidence about the antifungal activity of nanoparticles and evidence that this new alternative has more benefits when compared to current antifungal drugs available in the clinical setting.

Reviewer 1 ·

Basic reporting

In general, the manuscript's structure is correct and easy to follow.
In general, research is important, especially because it involves finding new alternatives in the treatment of pathogenic fungi. However, there are some suggestions that I propose to improve the work.

Experimental design

My main concern in the design of the experiment focuses on the antifungal ability of the synthesized compounds. Since it is one of the far-reaching objectives of said synthesis; I highly recommend posting some representative photographs of the antifungal activity of the synthesized NPs against the fungi that were evaluated.

Lines 33-346: These lines are of great interest to potential readers of the research and for this reason, I suggest microphotographs of the fungi tested against the nanoparticles, which could show their morphologies with or without the synthesized compounds.

Validity of the findings

I consider that the conclusion is not entirely supported by the results; More evidence is needed to conclude that they have antifungal effects.

Fig. 12 helps a little to understand but I consider that information is missing; For example, the NP bars are not observed for all fungi, nor with the bimetallic NPs.
Complete this figure, since there is no statistical analysis showing the significant differences. What is control, is it the extract only?

Additional comments

In the abstract section, as well as in the introductory section and the others; After you have written the full name of the microorganisms, you can abbreviate the genus.

In the methods section: ml --> mL and change rpm --> x g

Fig 1. Complete the legend for the Y axis, and the spectrum is missing for the extract as a control.

Reviewer 2 ·

Basic reporting

The description of results is ambiguous and needs improved English language used. Figures need to be changed or completed, labeled, and best described.

Experimental design

The research question was defined, in lines 33-34 this information is clear, but these results need to be compared with at least some treatments currently used for the pathogens that are mentioned. This information can be added in method point 2.7 lines 176-183 and demonstrated in Results point 3.4.
Methods used can be to replicate.

Validity of the findings

This information is a novelty, but the authors need to add more information by showing the benefit that using these nanoparticles compared with treatments used today.

Additional comments

The research “Bimetallic Nanoparticles and Biochar produced by Adansonia Digitata shell and their Effect against Tomato Pathogenic Fungi “proposes an alternative for new treatments against fungi, however, it does not include a comparison with currently used treatments.
Authors are advised to take into account the following observations.

• See in method UV-Vis does not show which is the equipment used to make this evaluation, while the other methods of characterization are explained. See lines 159-161.

• Review the punctuation marks in all text, for example, see line 44

• Read the information in 109-115 and try to change or improve this context because this text is similar to Results 189-194

• Improve graphic Figure 1, does not show “the label” in the vertical graph.

• All signals or FTIR, showed the functional groups of plant extract, while text describes that this behavior is because "capped with the secondary metabolites from the A. digitata fruit shell extract." In Harshiny Muthukumar, explain better how this signal can be improved https://doi.org/10.1016/j.btre.2020.e00469.

• sizes in figures 7 to 11 are not clear, is not possible to identify the size particles and identify the information that is described.

• ¿Figure 12 They labeled BiometallicsNPs and showed the activity in green color, why?

---

## Round 0.2 · Major Revisions

I thank the authors for the revised version of the manuscript. Most of the concerns raised by the Reviewers were addressed by the authors in this version. However, one essential aspect already pointed out by the reviewers is not included, the comparison of these proposed nanoparticles with current antifungal treatments for these pathogens. This aspect should be covered to place in perspective the real value of nanoparticles in the practical scenario.

---

## Round 0.3 · accepted · Accept

The authors addressed the pending issue related to comparison with antifungal drugs.